# Development and assessment of an immobilized bacterial alliance that efficiently degrades tylosin in wastewater

Boyu Zhao[1], Ye Wang[1], Jingyi Zhang[1], Lixia Wang[2], Wangdui Basang[3], Yanbin Zhu[3], Yunhang Gao[1]*

1 College of Animal Science and Technology, Jilin Agricultural University, Changchun, Jilin, China,
2 Northeast Institute of Geography and Agroecology, Chinese Academy of Sciences, Changchun, Jilin, China, 3 Institute of Animal Husbandry and Veterinary Medicine, Tibet Academy of Agricultural and Animal Husbandry Science, Lhasa, China

* gaoyunhang@163.com

**Data Availability Statement:** All relevant data are within the manuscript and its Supporting Information files.

## Abstract

Microbial degradation of tylosin (TYL) is a safe and environmentally friendly technology for remediating environmental pollution. Kurthia gibsonii (TYL-A1) and Klebsiella pneumonia (TYL-B2) were isolated from wastewater; degradation efficiency of the two strains combined was significantly greater than either alone and resulted in degradation products that were less toxic than TYL. With Polyvinyl alcohol (PVA)—sodium alginate (SA)—activated carbon (AC) used to form a bacterial immobilization carrier, the immobilized bacterial alliance reached 95.9% degradation efficiency in 1 d and could be reused for four cycles, with > 93% degradation efficiency per cycle. In a wastewater application, the immobilized bacterial alliance degraded 67.0% TYL in 9 d. There were significant advantages for the immobilized bacterial alliance at pH 5 or 9, with 20 or 40 g/L NaCl, or with 10 or 50 mg/L doxycycline. In summary, in this study, a bacterial consortium with TYL degradation ability was constructed using PVA-SA-AC as an immobilized carrier, and the application effect was evaluated on farm wastewater with a view to providing application guidance in environmental remediation.

## 1. Introduction

The agricultural environment is the sum of all elements of the surrounding environment with crops or agricultural production as the main body, including agricultural land, water use, atmosphere, and biology. Medical and agricultural industries widely use antibiotics to prevent and treat bacterial infections and improve growth [1]. For example, tetracyclines, macrolides, fluoroquinolones, and sulfonamides are commonly given to livestock [2]. However, they are not fully metabolized, resulting in high concentrations of parent compounds and metabolites in manure and urine [3]. Consequently, when such livestock manure is returned to the field, it may cause biotoxicity and promote development of antibiotic-resistant bacteria (ARBs) [4]. The accumulated toxicity of antibiotics and their side effects (allergenic, carcinogenic, teratogenic, and mutagenic) not only endanger the health of humans and animals [5], but also

**Funding:** The authors are thankful for the support of the Strategic Priority Research Program of the Chinese Academy of Science (XDA28080400), the Science and Technology program of Tibet Autonomous Region (XZ202101ZD0002N-04), and China Agriculture Research System of MOF and MARA (CARS-37).The funders had no role in study design, data collection and analysis, decision to publish, or preparation of the manuscript.

**Competing interests:** The authors have declared that no competing interests exist.

induce formation of antibiotic resistance genes (ARGs) and ARBs, and even superbugs (bacteria with resistance to multiple antibiotics), which in turn cause environmental pollution [6]. The presence of ARGs in soil, fruits, and vegetables were linked to livestock manure [7–9]. Tylosin (TYL) is produced by *Streptomyces* sp and belongs to the macrolide family [10]. It has a wide antibacterial spectrum, with a strong inhibitory effect on Gram-positive bacteria [11]. Concentrations of TYL in manure were 104.66 μg/kg for broilers [12], 0.22–0.28 mg/kg for dairy cattle, and 0.23–1.88 mg/kg for swine [13]. As these concentrations of TYL represent an environmental hazard, there is an impetus to explore effective degradation of TYL.

Researchers and industries have become increasingly interested in bioremediation using microorganisms as the need for greener solutions has grown over time [14]. In the environment, antibiotics are mainly degraded by microorganisms. Microbial degradation refers to the process of changing the structure and physicochemical properties of antibiotics by assimilation, with the large molecular structure of antibiotics degraded into small harmless molecules through a series of reactions [15]. It is noteworthy that ARBs have an important role in this process. For example, a strain of *Bacillus cereus* H38 caused 100% degradation of sulfamethazine (SMZ) at 25°C after 3 d [16]. Furthermore, *Klebsiella pneumoniae* TR5 degraded 90% of 200 mg/L tetracycline (TC) within 4 d [17]. *Bacillus* sp. degraded 75% of 25 mg/L TYL [18] and *Klebsiella* sp. degraded 99.3% of 25 mg/L TYL [19]. In addition, cultured microbial consortia degraded target contaminants better than single bacteria [20] and were a more effective bio-stimulation strategy [21]. Previous studies have shown that the degradation efficiency of bacterial consortia is higher than that of single bacteria [22–24]. Therefore bacterial co-cultures have good potential for degrading antibiotics under field conditions [25].

As microorganisms can be affected by unfavorable environments, functional microorganisms can be immobilized on carriers [26], thereby reducing external environment influences [27, 28]. Various materials are used as immobilization carriers, including inorganic materials, natural organic polymers, and synthetic organic polymers [29]. A popular material is PVA-SA (Polyvinyl Alcohol-Sodium Alginate), which is cost-effective and has high mechanical strength [30]. The use of PVA-SA to immobilize Bacillus cereus MRR2 removed 90.1% phosphate, 95.6% magnesium, and 95.7% ammonium ions [31], removing 75.52% of petroleum hydrocarbons [32]. In addition, PVA-SA can be used in combination with other materials, such as PVA-SA combined with biochar to establish a strain N80 immobilization system that degraded 89.37% TSM within 48 h [33]. Activated carbon (AC) is an excellent adsorbent with stable chemical properties, multiple pore sizes, large specific surface area, and abundant functional groups [5, 34]. Zhang et al. used micron-sized AC to immobilize diesel-degrading bacteria and removed 86.35% diesel in 15 d [35].

In this study, TYL was degraded for the first time by immobilized bacteria using PVA-SA-AC. The main objective of this study was to develop and evaluate a new physisorption and biodegradation system with the aim of efficiently removing TYL from the agro-environment in a green and economically sustainable way.

## 2. Materials and methods

### 2.1 Chemicals and media

The TYL (high-performance liquid chromatography (HPLC) grade $\geq$ 93%, CAS:1401-69-0) was purchased from Shanghai Yanaye Bio-Technology Co., Ltd (Shanghai, China), acetonitrile (HPLC grade; 99%; CAS:75-05-8) and methanol (HPLC grade; 99%; CAS: 67-56-1) from Thermo Fisher Scientific (Shanghai, China), formic acid (analytical purity grade; 95.5%; CAS: 64-18-6) from FUCHEN Chemistry (Tianjin, China). Polyvinyl alcohol 1788 (PVA; CAS: 9002-89-5) and sodium alginate (SA; CAS: 9005-38-3) were from Aladdin (Beijing, China). Mineral

salt medium (MSM; g/L) was comprised of: $MgSO_4$ 0.2 g, $KH_2PO_4$ 0.5 g, $K_2HPO_4$ 1.5 g, NaCl 1 g, and yeast extract 1.0 g. Composition of the Luria-Bertani medium (LB; g/L) was: tryptone 10.0 g, NaCl 10.0 g, and yeast extract 5.0 g. The TSA and TSB were from Qingdao Hope Bio-Technology Co., Ltd (Qingdao, China). Unless specified, all other chemicals used in this study were analytical purity grade.

## 2.2 Isolation and identification of TYL-degrading bacteria

Wastewater samples were collected from Guangze ecological pasture (Changchun, China). Soil samples were collected from Jilin Agricultural University (Jilin, China), with 2 g samples added to 0.85% saline with glass beads in a shaker at 37°C for 2 h and allowed to stand for 1 h. Then, 1 mL supernatant was added to MSM containing 20 mg/L TYL. Gradient domestication was performed for four cycles, and TYL concentrations were 20 to 50 mg/L. The supernatant was taken and spread on an MSM plate containing 50 mg/L TYL, and single colonies were picked and streaked on a TSA plate until single colonies were separated. Genomic DNA of the strains was extracted using the Bacterial Genomic DNA Isolation Kit (Sangon, Shanghai, China). The 16S rRNA gene was amplified by polymerase chain reaction (PCR) using universal primers 27F (5′–AGAGAGTTTGATTGGCTCAG–3′) and 1492R (5′–GGTTACCTTTGTTACGACTT–3′), and PCR products were sequenced by Sangon Biotechnology (Shanghai, China). The 16S rRNA sequences were compared to the NCBI GenBank database via online BLAST and the sequence with a high alignment score was selected to construct the phylogenetic tree with MEGA 7.0.

## 2.3 Compatibility of strain TYL-A1 with strain TYL-B2 and assessing bacterial alliance degradation

An Oxford cup inhibition circle experiment was done to detect antagonism between the two strains. First, 200 μL of coated TYL-A1 bacterial solution was spread evenly on the TSA plate. Then, a sterile Oxford cup was placed on the plate, 200 μL TYL-B2 bacterial solution added to the Oxford cup (or sterile TSB as control), followed by incubation for 24 h to allow formation of an inhibition circle. The two strains were cultured to logarithmic phases and the cultures were centrifuged at 8500 rpm for 15 min at 4°C. The supernatant was discarded, the bacteria were washed three times with PBS, and $OD_{600}$ was adjusted to 2.0, and the bacterial suspension was completed. The two strains were added to MSM at a ratio of 1:1 and 5% inoculum for the TYL degradation experiment. Degradation experiments were performed in 100 ml of MSM containing 75 mg/L TYL. The TYL concentrations were measured by HPLC Shimadzu LC-2030 Plus (Kyoto, Japan). The following chromatographic conditions were used: Column C18 (chromatographic column was an Agilent ZORBAX SB-C18 (250 mm × 4.6 mm, 5 μm, Santa Clara, CA, USA)), The mobile phase consisted of 0.01% formic acid and acetonitrile (67:33); the mobile phase flow rate was 0.6 mL/min; the column temperature was 40°C, and the injection volume was 15 μL. Degradation efficiency (%) was calculated as follows:

$$\frac{C0 - Ct}{C0} \times 100\%$$

Where $C_0$ is the initial concentration of TYL (mg/L), and $C_t$ is the residual concentration of TYL in the solution (mg/L).

## 2.4 Ecotoxicity assessment of TYL products degraded by the bacterial alliance

The $OD_{600}$ of *Escherichia coli* (ATCC 25922) and *Staphylococcus aureus* (ATCC 25923) (stored in the laboratory) were used as indicators for bio-toxicity assessment. Both *E. coli* and *S. aureus*

were resuscitated and cultured in LB broth. There were three treatments: (T) bacterial alliance with 150 mg/L of TYL; (PC) no bacteria and TYL; and (NC) 150 mg/L of TYL only. All cultures were incubated at 35˚C in the dark at a shaking speed of 135 rpm for 12 h. The cultures were centrifuged at 4˚C for 10 min at 8000rpm and the supernatant filtered through a 0.22 μm membrane (0.22 μm polytetrafluoroethylene filters, Jinteng, Tianjin, China). Then, *E. coli* and *S. aureus* were added to the three groups, respectively, and absorbance at 600 nm was measured using a UV/visible spectrophotometer (MU701, Shimadzu, Kyoto, Japan).

## 2.5 Preparing the immobilized bacterial alliance and assessing degradation performance

Activated carbon (AC) was washed three times with deionized water and dried at 65˚C. The bacterial alliance was immobilized with 2% SA+2% PVA (SA and PVA ratios were based on [26] with various concentrations of AC (0, 0.5, 1, 1.5, and 2%). First, PVA was dissolved by heating and SA added, with mixing by a magnetic stirrer. Finally, AC was added, allowed to dissolve and time allowed to eliminate air bubbles. The bacterial alliance and the PVA-SA-AC carrier mixture (v:v = 1:5) were added in drops to the autoclaved 2% calcium chloride saturated boric acid solution (2 g calcium chloride, 3.9 g boric acid in 100 mL ultrapure water) and kept at 4˚C for 24 h to allow full calcification into beads. Then, the microspheres were taken out, washed with PBS, blotted with filter paper, and stored at 4˚C. Thereafter, 5 g beads were added to 100 mL of ultrapure water and shaken for 15 d at 35˚C, 180 rpm/min to induce fragmentation and determine mechanical strength of the beads. In addition, bead stability was assessed by immersing them in acid, alkali, and salt solutions. Based on the above experiments, the construct of the carrier was completed, and the bacteria-free carrier was assessed by scanning electron microscopy (SEM). Finally, the free bacterial alliance and immobilized beads were added to the MSM containing TYL, and degradation efficiency of the immobilized bacterial alliance was evaluated.

## 2.6 Reusability and protective performance of the immobilized bacterial alliance

To test bead reusability, four degradation experiments were performed in the same mode, for 48-h intervals. Immobilized beads were removed at the end of each degradation experiment, washed repeatedly with sterile distilled water, and then placed in MSM containing TYL for degradation. Bead reusability was assessed by degradation efficiency.

To determine protective effects of immobilization, the free bacterial alliance and immobilized bacterial alliance were added into MSM with various conditions: pH 5, 7, or 9; NaCl concentrations of 20 or 40 g/L; 10 or 50 mg/L doxycycline (DOX); and MSM without free and immobilized bacteria alliance as control. Immobilization protection was assessed by degradation efficiency.

## 2.7 Assessing TYL degradation capacity of the immobilized bacterial alliance under farming wastewater conditions

Degradation application experiments were conducted to explore performance of immobilized bacterial alliance in wastewater containing TYL. The immobilized beads were added to 100 ml wastewater containing TYL. Samples were collected on days 1, 3, 5, 7, and 9, and concentrations of TYL, COD, $NH_4^+-N$, $NO_3^--N$, and $PO_4^{3-}-P$ were measured. The COD was measured using the potassium dichromate method (HJ 535–2007), $NH_4^+-N$ by the spectrophotometric method of nano reagent (HJ 535–2009), $NO_3^--N$ by the method of color-changing acid, and $PO_4^{3-}-P$ by the spectrophotometric method of ammonium molybdate (GB 11893–89).

## 2.8 Data and statistical analyses

Unless otherwise specified, TYL concentration in the degradation experiment was 150 mg/L and the conditions were 35°C, 135 rpm, 4 d, conducted in triplicate, and no bacteria as control. Data were analyzed by one-way ANOVA using SPSS software (Version 25.0) and graphs prepared using GraphPad Prism (Version 7.0).

# 3. Results and discussion

## 3.1 Identification of strains TYL-A1 and TYL-B2, construction and degradation performance of the bacterial alliance

After four rounds of domestication and enrichment, two suitable strains were selected and designated strains TYL-A1 and TYL-B2 derived from soil and wastewater, respectively. On TSA plates, colonies of TYL-A1 were irregular, yellowish, opaque, and had irregular edges, whereas colonies of TYL-B2 were round, white, opaque, and with well-defined edges. Based on the phylogenetic tree, strain TYI-A1 (GenBank OP077323.1) was closest to *Kurthia gibsonii* with 99.86% similarity (Fig 1A). To our knowledge, TYL degradation by *Kurthia gibsonii* has not been reported. Strain TYL-B2 (GenBank OP077332.1) was closest to *Klebsiella pneumoniae* with 99.14% similarity (Fig 1B). Strain TYL-A1 had logarithmic phase growth in 2–20 h, with growth stabilizing after 20 h (Fig 1C). The logarithmic phase of TYL-B2 was 2–16 h, and the stabilization period was 16–24 h (Fig 1D).

Results of the Oxford cup inhibition circle are in S1 Fig. As no inhibition circle appeared around the Oxford cup, the two strains were not antagonistic. Degradation of TYL by strains

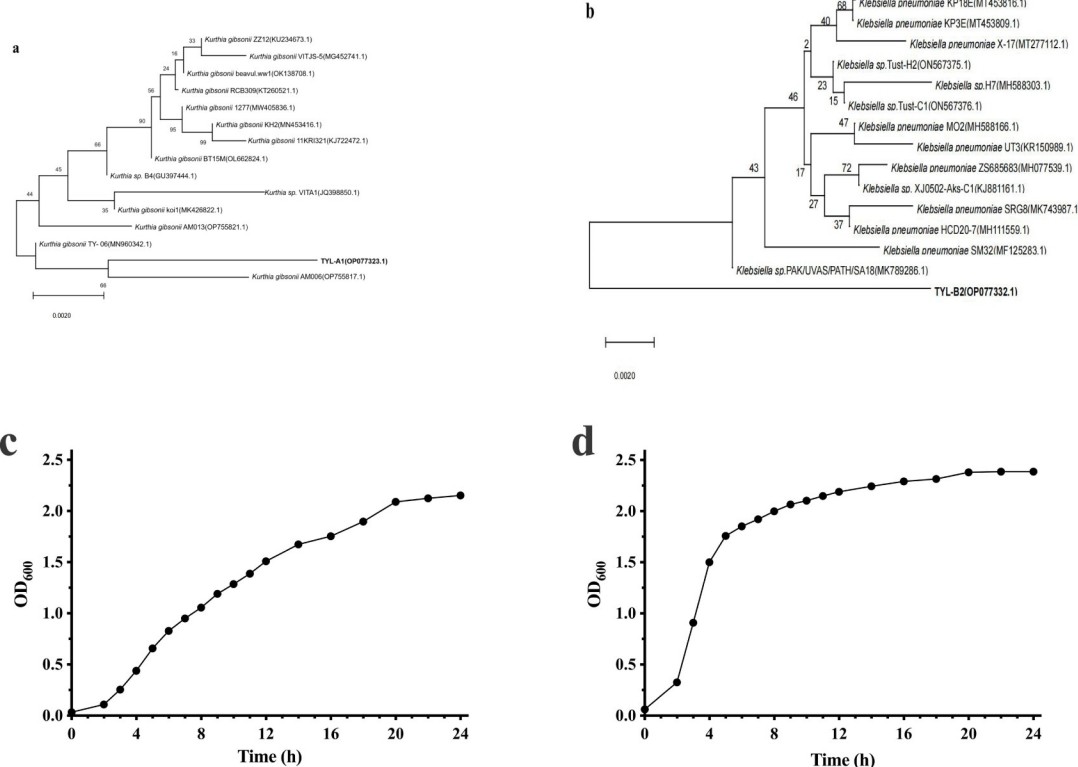

**Fig 1.** Phylogenetic tree (neighbor-joining method) and growth curve of strain TYL-A1 and TYL-B2, **a**, **c**) strain TYL-A1 and **b**, **d**) strain TYL-B2.

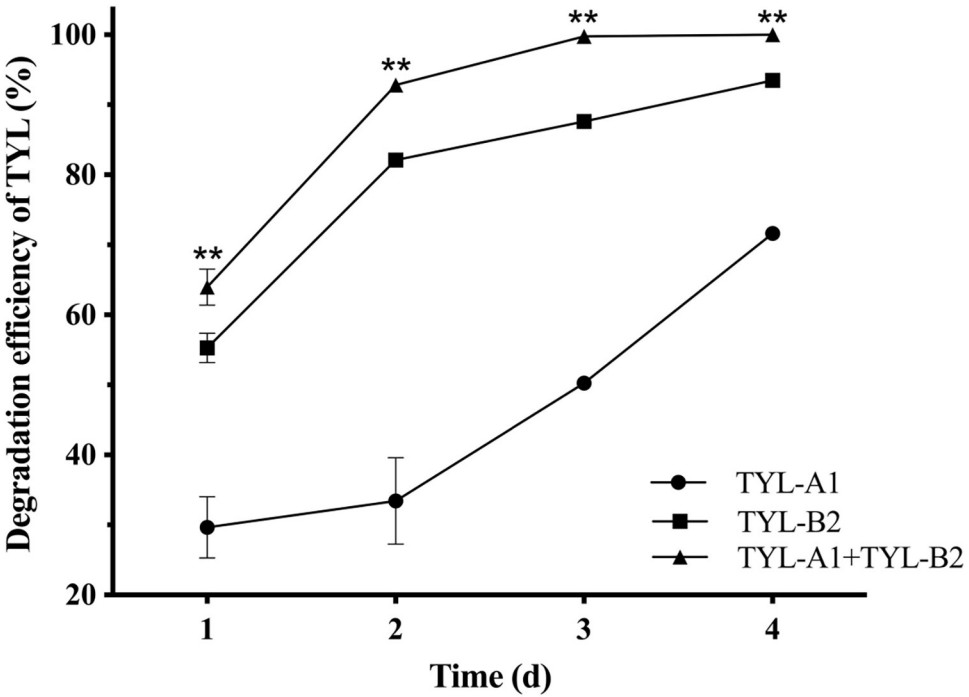

**Fig 2. TYL degradation efficiency under TYL-A1, TYL-B2, and bacterial alliance.** Data points are means and bars represent SEM of three experiments: ** ($p < 0.01$).

TYL-A1, TYL-B2, and bacterial alliance are shown in Fig 2. Strain TYL-B2 degraded 93.5% TYL and strain TYL-A1 degraded 71.6% TYL in 4 d. However, bacterial alliance increased degradation efficiency of TYL ($p < 0.01$), degrading 100% TYL at 4 d.

Co-cultured bacterial alliances degrade contaminants better than single bacteria, perhaps due to metabolite exchanges between the two strains [21]. In this study, two TYL-degrading bacteria were isolated and combined to make a bacterial alliance that had higher degradation efficiency than only one strain. Previous studies used bacterial consortia to degrade ciprofloxacin [22], terramycin [36], and tiamulin (TIA) [37]. However, the present study was apparently the first to use a bacterial alliance to degrade TYL. Co-existing bacteria can benefit from the division of labor [38],with metabolites released by one bacterial strain used as a nutrient by another [39]. Furthermore, synergism between bacteria, increased degradation genes, and enzymes necessary for complete degradation will also increase degradation capacity of bacterial alliances [40].

### 3.2 Biotoxicity of TYL degradation products

Although the constructed bacterial alliance was effective in degrading TYL, ecotoxicity of the degradation products was unknown. Therefore, its biotoxicity was explored. The $OD_{600}$ of the indicator bacteria in the T group treatment was higher than the NC group and lower than PC group (Fig 3), indicating that toxicity of the degradation products was lower than TYL, with residual weak biotoxicity. In previous studies, degradation products of CTC by strain LZ01 and OTC by strain OTC-16 were less toxic than the parent compound [41, 42].

### 3.3 Features of immobilized bacterial alliance

Effect of different concentrations of AC on beads (S2 Fig), increasing AC concentration did not affect particle shape, but 1.5 and 2% AC decreased mechanical strength and breakage occurred

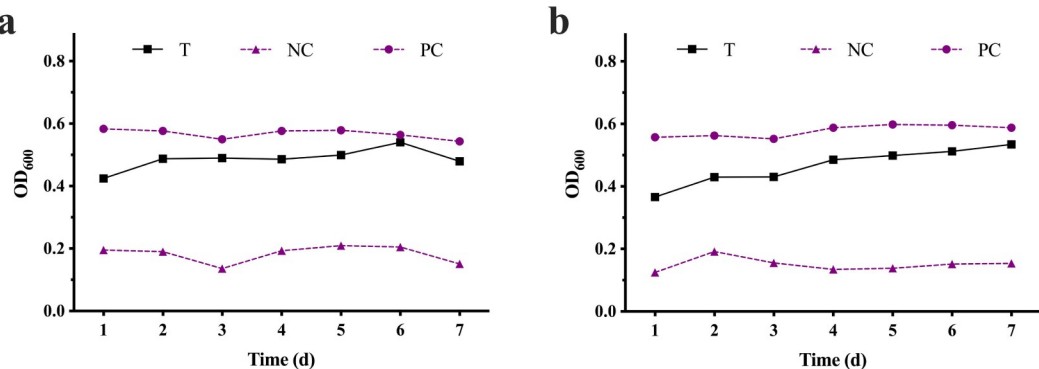

**Fig 3.** Bacterial inhibition of **a**) *Escherichia coli* and **b**) *Staphylococcus aureus* by bacterial alliance degradation TYL products. (T) bacterial alliance with 150 mg/L of TYL, (PC) no bacteria and TYL, and (NC) 150 mg/L of TYL only.

on the third day. Unadded AC microspheres slowly became transparent and had volume expansion in acid and alkali solutions and appeared to dissolve after 7 d (Table 1). However, bead stability was increased by adding 0.5 or 1% AC. Increased AC concentrations can increase mechanical strength of beads, but excessive AC can reduce bead stability. Therefore, 0.5% AC was used.

Degradation of TYL by the free bacterial alliance, blank beads, and immobilized bacterial alliance are shown (Fig 4). In the first 2 d, degradation by immobilized beads was higher than free bacterial alliance and blank beads ($p < 0.01$), and degradation efficiency reached 99.2% on the fourth day. Although degradation efficiency of blank beads reached 30.4% on the fourth day, this was attributed to AC causing adsorption of TYL but not degradation [43]. Degradation has two phases. First, there is adsorption of TYL; the immobilized carrier provides protection for growth of the bacterial alliance and increases effective contact area between the immobilized bacterial alliance and TYL [44]. Thereafter, biodegradation occurred with the bacterial alliance embedded in the carrier using TYL as a carbon source. Both biodegradation and physical adsorption acted synergistically to accelerate TYL degradation, in contrast to inefficient degradation by free bacteria. Similarly, cyanide was removed from polyurethane foam immobilized *Alcaligenes sp* [45] and aromatic compounds were removed by adsorptive silica encapsulated *Pseudomonas sp* [46] through adsorption and biodegradation. However, this is apparently the first report of TYL degradation with PVA-SA-AC immobilized bacteria, with potential for environmental decontamination.

Based on SEM(Scanning Electron Microscope) imaging, there were numerous voids in blank beads (Fig 5A), and bacteria adhered to these voids (Fig 5B).Furthermore, in immobilized carriers, these voids also transferred TYL to the interior, accelerating its degradation by the

**Table 1. Stability of microspheres in acid, alkali, and salt solutions.**

| Content of AC | Acid | Base | Salt |
|---|---|---|---|
| 0% | ++++ | +++ | ++ |
| 0.50% | ++ | ++ | + |
| 1% | ++ | ++ | + |

+ slightly softened, slightly dissolved

++ softened, dissolved

+++ softened, more dissolved, loose structure

++++ softened and dissolved more, severely deformed.

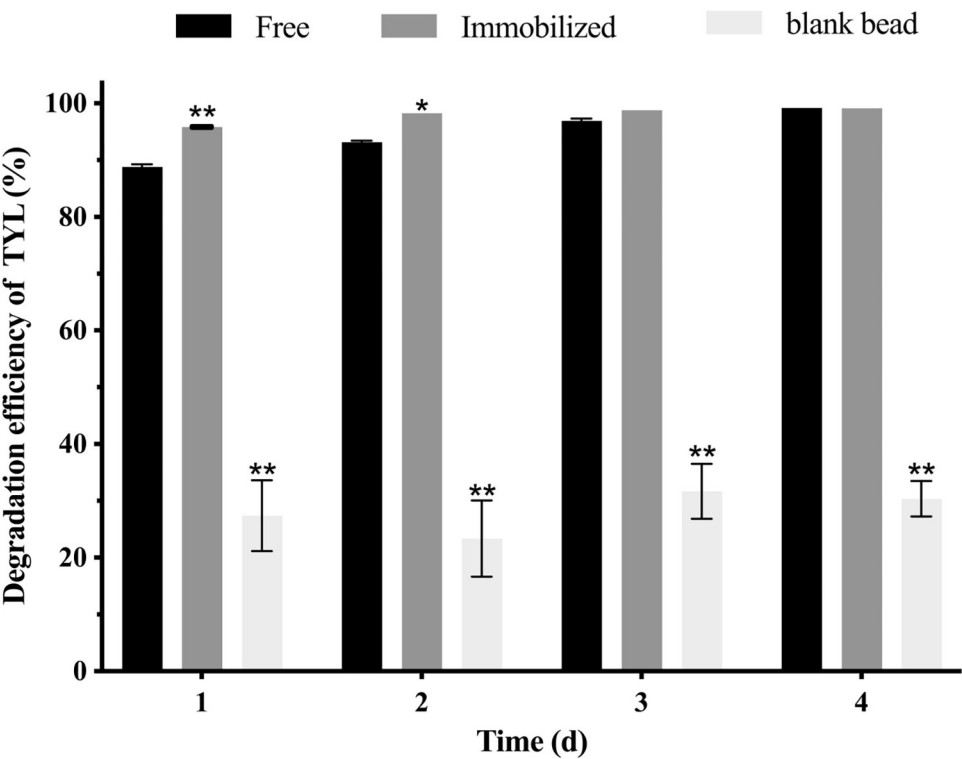

**Fig 4. TYL degradation efficiency under free bacteria concretization, immobilized bacteria concretization, and blank bead.** Data points are means and bars are SEM of three experiments; * $(0.01 < p < 0.05)$; ** $(p < 0.01)$.

bacterial alliance. Adsorption by immobilized carriers and biodegradation by the bacterial alliance synergistically worked together to promote and sustain TYL degradation [35] In short, immobilized carriers enabled physical adsorption and biodegradation to promote TYL degradation.

## 3.4 Protective effects of immobilization and reusability of immobilized bacterial alliance

Adverse conditions can inhibit bacterial growth [47, 48], but immobilization can confer protection [49]. Therefore, protective effects of immobilized carriers on the bacterial alliance at

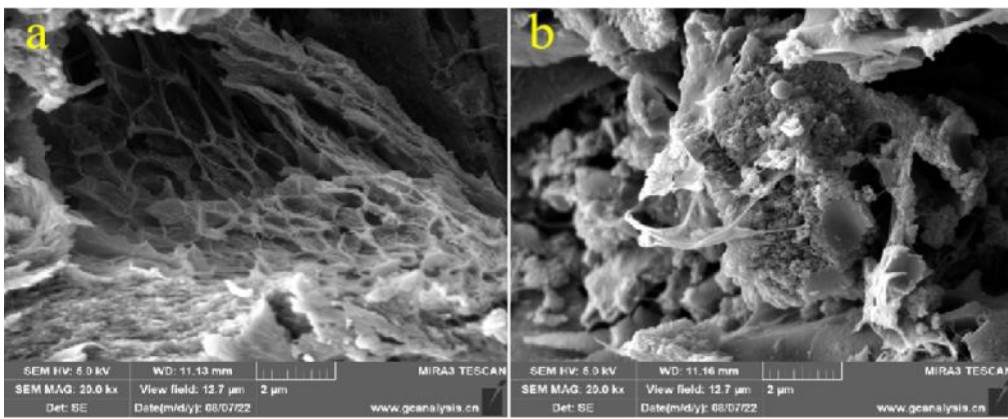

**Fig 5.** SEM images of **a**) blank beads and **b**) immobilized bacterial alliance.

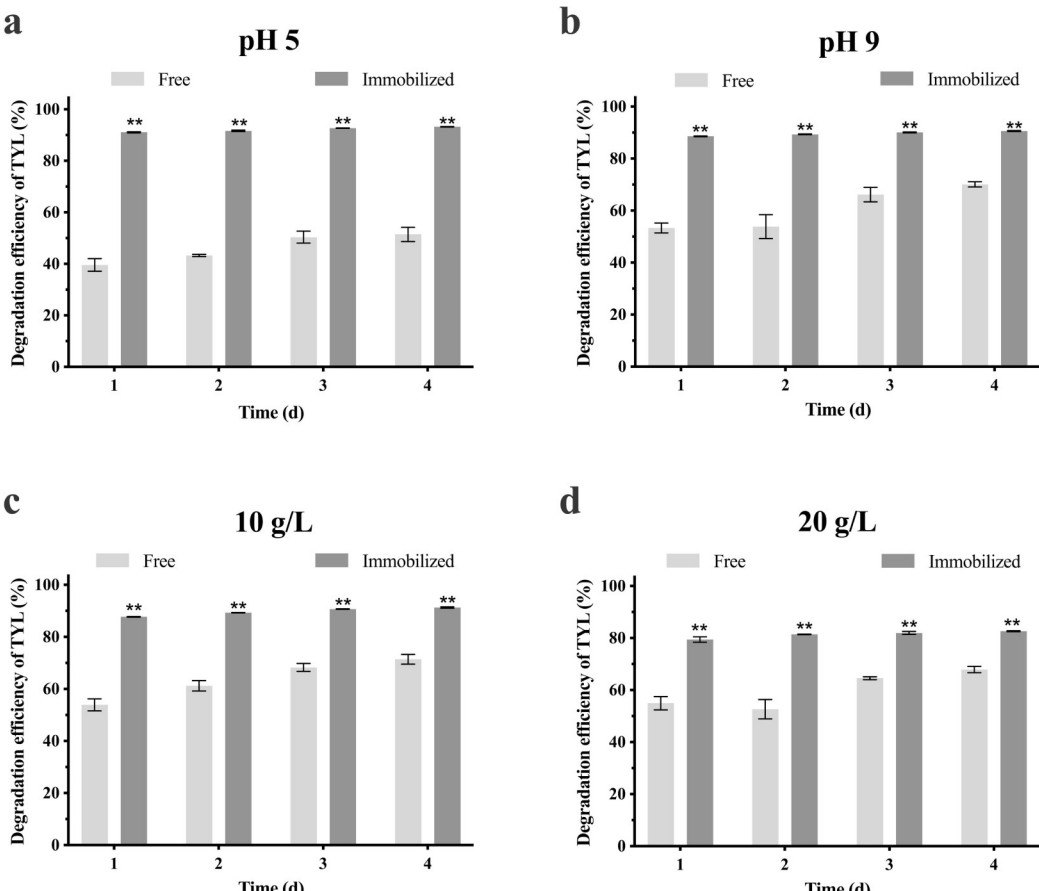

**Fig 6.** Degradation efficiency of TYL by immobilized bacterial alliance and free bacterial alliance under various conditions; **a**) pH 5, **b**) pH 9, **c**) 20 g/L NaCl, and **d**) 40 g/L NaCl. Data points are means and bars represent SEM of three experiments: ** ($p < 0.01$).

various pH and salt concentrations was evaluated (Fig 6). Degradation of TYL by the free bacterial alliance was affected by all four conditions, although effects of salt concentration were less than the pH. At 10 g/L NaCl, the free bacterial alliance degraded 71.4% TYL in 4 d, and the immobilized bacterial alliance increased degradation efficiency to 91.3% ($p < 0.01$). At 20 g/L NaCl, degradation of the free bacterial alliance was inhibited, with only 67.9% degraded by the fourth day. After immobilization, degradation efficiency was also reduced but still higher than the free bacteria alliance (82.6%).A high salt concentration results in plasmolysis of bacterial cells, and bacterial growth is reduced [50] resulting in lower degradation rates.pH is one of the most important parameters affecting the activity of enzyme and degradation potential of the bacteria [51]. Regarding pH, the free bacterial alliance degraded better under alkaline conditions. After immobilization, degradation efficiency reached 90.6%, consistent with previous studies that immobilization can improve degradation efficiency of bacteria in adverse environments [52, 53]. Increased degradation efficiency may be due to higher tolerance of an immobilized bacterial alliance to environmental changes, as described [30, 54].

Reusability is an advantage of immobilization, with bead recycling increasing sustainability. The immobilized bacterial alliance was reused for four cycles (Fig 7), with a 93.6% degradation efficiency of TYL in the fourth cycle. Microspheres started to break up after four cycles, although the beads did not break after 15 d of shaking at 180 rpm in MSM during the

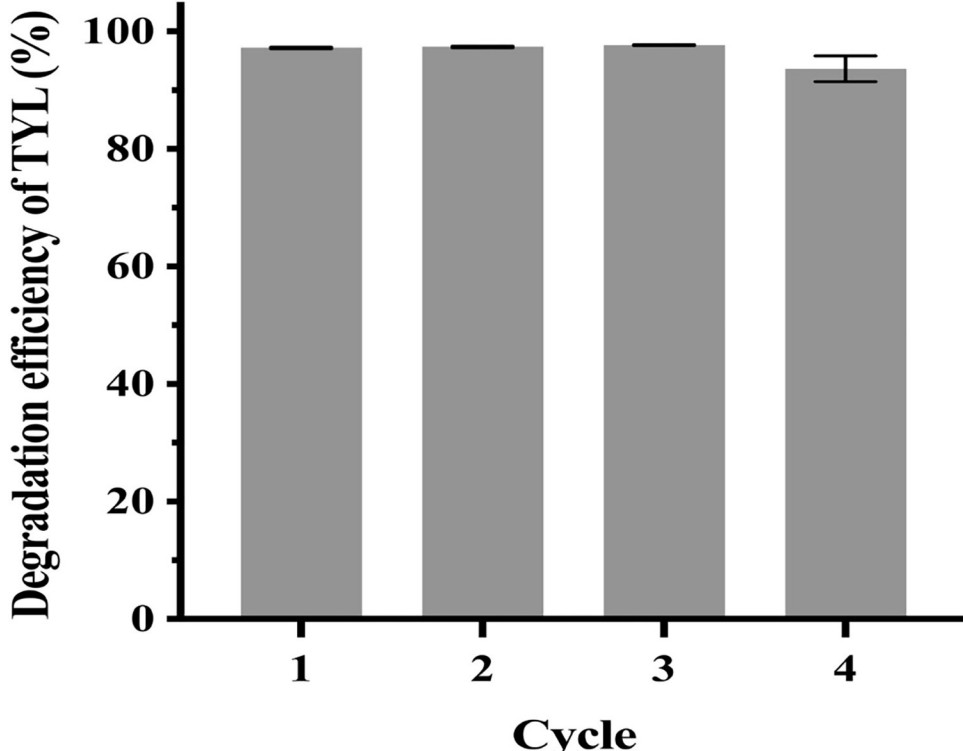

**Fig 7. Degradation efficiency of TYL by immobilized bacterial alliance per cycle.** Data points are means and bars represent SEM of three experiments.

mechanical stability test. Breakage may be accelerated by the long-term exposure to solutions containing organic contaminants and undergoing frequent environmental changes, and by growth of the immobilized bacterial alliance. Immobilization protected the bacteria in adverse environments and beads could be recycled up to four times. Iron-oxide nanoparticles to immobilize *Bacillus subtilis* by adsorption could be recycled 7 times [55],whereas immobilized *Serratia marcecens* using biochar, PVA and SA appeared broken in the 5th cycle [33]. Perhaps additional substances can increase the strength of the immobilized carriers to adapt to more complex and variable environments.

## 3.5 Degradation of TYL by immobilized bacterial alliance in presence of DOX

As TYL is often used in combination with DOX, degradation ability of the immobilized bacterial alliance against TYL was determined under variable DOX concentrations. At 10 and 50 mg/L DOX, the free bacterial alliance degraded 93.7 and 86.6% TYL, respectively, after 4 d (Fig 8), indicating that the bacterial alliance was resistant to DOX. Furthermore, the immobilized bacterial alliance degraded 98.8 and 97.3% TYL at 10 and 50 mg/L DOX, respectively. Therefore, immobilization reduced impacts of DOX on the bacterial alliance and ensured a stable degradation capacity. Although the immobilized system improved the adaptation and resistance of bacteria, degradation efficiency was still reduced when compared to no DOX (Fig 4). Potential reasons include: DOX inhibited growth of bacteria; the immobilized carrier first adsorbed DOX, which inhibited TYL delivery to the inner part of the beads; and carbon catabolite repression [56]. Bacteria preferentially adapt to carbon or energy sources that are rapidly

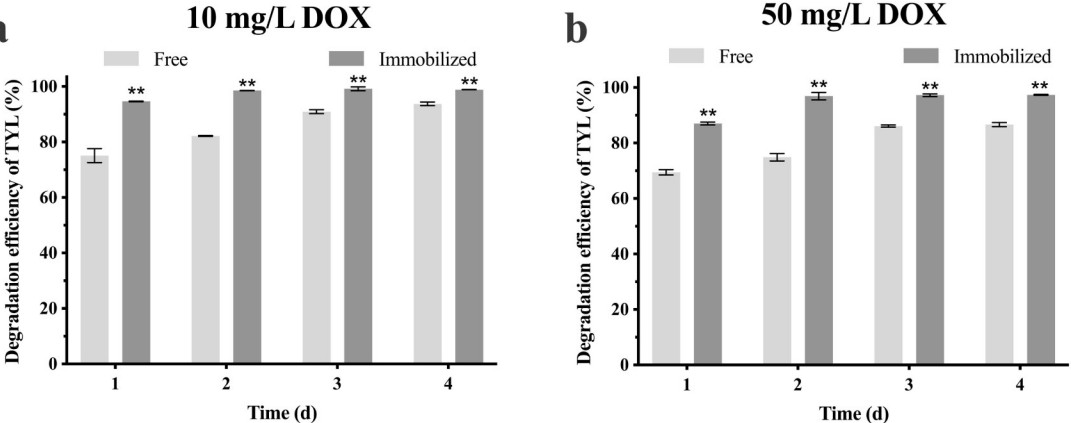

**Fig 8.** Degradation efficiency of TYL by immobilized bacterial alliance and free bacterial alliance under **a**) 10 mg/L DOX, **b**) 50 mg/L DOX. Data points are means average and bars represent SEM of three experiments: ** ($p < 0.01$).

biodegradable, whereas inhibiting enzymes use other carbon and energy sources. Dual substrates can provide additional energy and nutrients to microorganisms, but may also cause inhibition by carbon catabolite repression [57]. In this experiment, the bacterial alliance may have preferentially used DOX, and the presence of DOX promoted bacterial growth and reduced biodegradation of TYL. Degradation of phenanthrene and anthracene using ASPF (enriched mixed bacterial cultures) with the addition of intermediate products phthalic acid and catecho (preferred carbon sources for ASPF) reduced degradation efficiency of phenanthrene and anthracene [58]. Similar to the present study, TX-100 during degradation of PVA-SA immobilized *Mycobacterium crocinum* strain NJS-1 inhibited degradation efficiency of phenanthrene [27]. The presence of high versus low DOX concentrations may have promoted proliferation of the bacterial alliance, but also reduced degradation of TYL.

### 3.6 Degradation of TYL in wastewater by the immobilized bacterial alliance

Although the immobilized bacterial consortium had excellent degradation of TYL under laboratory conditions, degradation ability in a field-type environment was unknown. In wastewater, TYL degradation efficiency increased with time, reaching 67% on the ninth day (Fig 9A), lower than in MSM (Fig 4). There are two potential explanations. First, adsorption capacity of immobilized beads is limited and other substances in the wastewater may preferentially occupy adsorption sites. Secondly, as wastewater from aquaculture is rich in organic substances [59], with some potentially inhibiting growth of bacteria. Similarly, in previous study, TYL-Y13 degradation efficiency was reduced in swine wastewater, degrading 30% of TYL by 84 h [60]. Although the degradation efficiency of immobilized strains is low, the immobilized beads can be recycled to save energy.

In this study, COD, P, $NO_3^-$–N, and $NH_4^+$–N were measured during degradation. The COD of the experimental group increased and then decreased (Fig 9B). The initial increase was attributed to TYL and immobilized carriers, in the presence of TYL and immobilized carriers, there are too many composite carbon sources, leading to an increase in the concentration of organic matter and a corresponding increase in the value of COD. Too many composite carbon sources also lead to an accelerated rate of metabolism of microorganisms, and the biological decomposition of organic matter in the wastewater is rapid, resulting in the production of more COD, leading to an increase in the content of COD.whereas the decrease was attributed to degradation of TYL and possibly other substances in the wastewater. On the ninth day, the

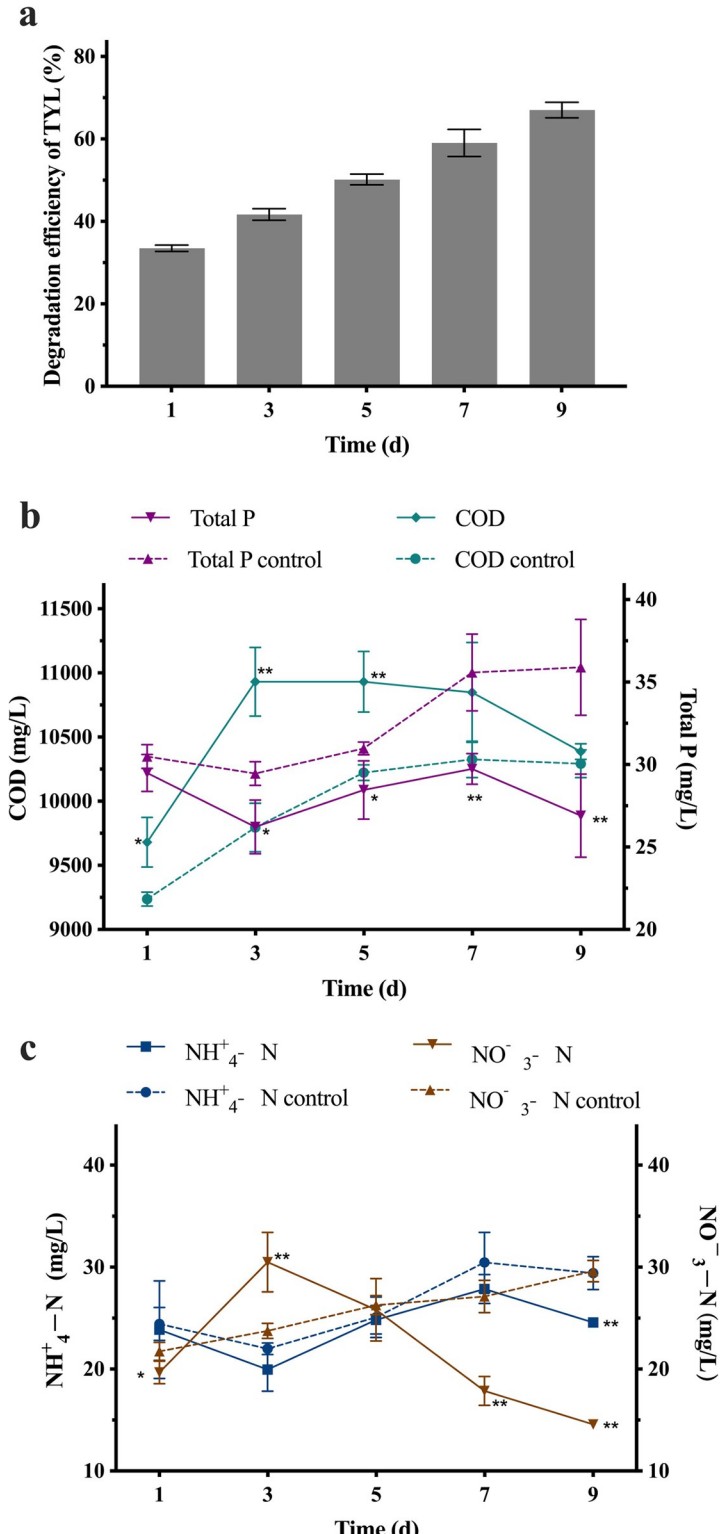

**Fig 9. Changes in TYL degradation efficiency, COD, total P, NH4+−N, and NO3—N in wastewater. a**) TYL degradation efficiency, **b**) COD and total P, **c**) $NH_4^+$−N and $NO_3^-$N. Data points are means and bars represent SEM of three experiments; * $(0.01 < p < 0.05)$; ** $(p < 0.01)$.

COD of the experimental group and the blank group were not significantly different. However, total P was significantly lower in the experimental group than in the control, probably due to bacteria utilizing P (Fig 9B). Both $NH_4^+-N$ and $NO_3^--N$ increased in the control group (Fig 9C), likely due to nitrification and denitrification reactions. After addition of immobilized beads, $NO_3^--N$ increased and then decreased, whereas $NH_4^+-N$ had opposite changes, attributed to enhanced nitrification reaction in the early stages and enhanced denitrification reaction in the later stages. On the ninth day, both $NO_3^--N$, and $NH_4^+-N$ were significantly lower than in the control group. In wastewater, immobilization sustained TYL degradation capacity of the bacterial alliance, with potential for application under field conditions.

## 4. Conclusions

Two degrading strains were isolated from the wastewater and identified as *Kurtosis giganteus* (TYL-A1) and *Klebsiella pneumonia* (TYL-B2). The two strains were constructed into a bacterial alliance. The TYL degradation capacity of the bacterial alliance was higher than either strain of bacteria and degradation products were less toxic than the parent compound. Next, bacterial immobilization carriers were constructed using PVA-SA-AC. The immobilized bacterial alliance removed TYL by a combination of biosorption and biodegradation, degrading 93.6% of TYL after four cycles. In adverse conditions (pH 5, pH 9, 20 g/L, 40 g/L NaCl, 10 mg/L, and 50 mg/L DOX), degradation efficiency of the immobilized bacterial alliance was significantly higher than the free bacterial alliance. Finally, the combination of immobilized bacteria applied to wastewater achieved 67% TYL degradation efficiency. In conclusion, abacterial alliance was constructed and immobilized; the immobilized bacterial alliance had much promise, particularly under field conditions. The mechanism of bacterial alliance degrading TYL and the combination of immobilized bacterial association with wastewater treatment equipment will be investigated in the future.

## Supporting information

**S1 Fig. Oxford cup results of TYL-A1 and TYL-B2.** 1, 2, 3 for the experimental group, 4 for the blank control.
(PDF)

**S2 Fig.** The characteristics of the beads with different AC contents, a)-e) were 0.5%, 1%, 1.5%, 2%, and 2.5%, respectively.
(PDF)

## Author Contributions

**Conceptualization:** Boyu Zhao, Ye Wang, Jingyi Zhang.

**Data curation:** Boyu Zhao, Ye Wang, Yunhang Gao.

**Formal analysis:** Boyu Zhao, Ye Wang.

**Funding acquisition:** Lixia Wang, Wangdui Basang, Yanbin Zhu, Yunhang Gao.

**Investigation:** Ye Wang.

**Methodology:** Boyu Zhao, Ye Wang, Jingyi Zhang, Yunhang Gao.

**Project administration:** Yunhang Gao.

**Resources:** Wangdui Basang.

**Software:** Boyu Zhao, Ye Wang.

**Supervision:** Jingyi Zhang.

**Validation:** Lixia Wang, Yanbin Zhu.

**Visualization:** Lixia Wang, Wangdui Basang, Yanbin Zhu, Yunhang Gao.

**Writing – original draft:** Boyu Zhao, Ye Wang.

**Writing – review & editing:** Yunhang Gao.

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
