## [Decision Letter · Decision Letter 0]

19 Feb 2024

PONE-D-23-42913Development and assessment of an immobilized bacterial alliance that efficiently degrades tylosin in wastewaterPLOS ONE

Dear Dr. Gao,

Thank you for submitting your manuscript to PLOS ONE. After careful consideration, we feel that it has merit but does not fully meet PLOS ONE’s publication criteria as it currently stands. Therefore, we invite you to submit a revised version of the manuscript that addresses the points raised during the review process.

Authors <word-sub class="revised" data-idx="0-0-0">are required</word-sub> to <word-sub class="revised" data-idx="0-0-1">assess</word-sub> and <word-sub class="revised" data-idx="0-0-2">implement</word-sub> the <word-sub class="revised" data-idx="0-0-3">essential revisions</word-sub> for a <word-sub class="revised" data-idx="0-0-4">subsequent</word-sub> round of <word-sub class="revised" data-idx="0-0-5">manuscript review.</word-sub>
<word-sub class="revised" data-idx="0-1-0">The reviewers raised several concerns, such as</word-sub> the biosafety of the bacterial <word-sub class="revised" data-idx="0-1-2">strain</word-sub> and the efficiency of pollutant <word-sub class="revised" data-idx="0-1-5">species degradation.</word-sub>

We look forward to receiving your revised manuscript.

Kind regards,

Marcos Pileggi, Ph.D

Academic Editor

PLOS ONE

Journal Requirements:

"Research Program of the Chinese Academy of Science (XDA28080400)

Technology program of Tibet Autonomous Region (XZ202101ZD0002N-04)

China Agriculture Research System of MOF and MARA (CARS-37)"

"The authors are thankful for the support of the Strategic Priority Research Program of the Chinese Academy of Science (XDA28080400), the Science and Technology program of Tibet Autonomous Region (XZ202101ZD0002N-04), and China Agriculture Research System of MOF and MARA (CARS-37)."

"Research Program of the Chinese Academy of Science (XDA28080400)

Technology program of Tibet Autonomous Region (XZ202101ZD0002N-04)

China Agriculture Research System of MOF and MARA (CARS-37)"

Reviewers' comments:

Reviewer's Responses to Questions

**Comments to the Author**

1. Is the manuscript technically sound, and do the data support the conclusions?

Reviewer #1: Partly

Reviewer #2: Yes

2. Has the statistical analysis been performed appropriately and rigorously? 

Reviewer #1: Yes

Reviewer #2: Yes

3. Have the authors made all data underlying the findings in their manuscript fully available?

Reviewer #1: No

Reviewer #2: Yes

4. Is the manuscript presented in an intelligible fashion and written in standard English?

Reviewer #1: Yes

Reviewer #2: No

5. Review Comments to the Author

Reviewer #1: In their work, Zhao et al examined the possibility of using a consortium consisting of two newly isolated bacterial strains, Kurthia gibsonii (TYL-A1) and Klebsiella pneumonia (TYL-B2), for microbial degradation of tylosin. For better performance, the consortium was immobilized in PVA-SA-AC. Possible mechanisms for improving TYL biodegradation through immobilization are discussed. The feasibility of TYL removal was tested using real wastewater. Despite the overall good impression of the paper, I have some concerns, mainly about the real wastewater part and the pathogenicity, see comments below:

Specific comments:

Line 18: “PVA-SA-AC” should be defined upon first use

Lines 74-77: The poor sentence construction, please check

Line 79: what is TSM?

Lines 86-87: “from an agricultural environment.” What is the agricultural environment? Please be more specific when formulating the purpose of the work.

Line 121: “spresd” please check

Line 147: “0.22 mm membrane” please check the units are correct.

Line 186: “in wastewater TYL” change to "in wastewater containing TYL"

Lines 186-187: “standing wastewater and supernatant were” please give more detail what is standing wastewater and supernatant?

Line 188: “wastewater containing TYL” how much TYL is contained in wastewater?

Line 189: “NH+ 4−N” please change to “NH4+-N”. Please use the same format for other nitrate-N and phosphate-P

Line 246: what are CTC and OTC?

Line 261: “++++ softened…” please check, should be +++

Line 266: What does the word “concretization” mean here?

Line 279: “in contrast to inefficient degradation by free bacteria” degradation by free bacteria was not inefficient

Line 293: "a" and "b" are not shown on images

Line 315: “represent SEM of three experiments” please define SEM here

Line 315: “wastewater from aquaculture” Wastewater from aquaculture or agriculture? please clarify

Line 373: “Degradation efficiency was reduced, but it can be recycled.” the meaning is not clear, please rephrase

Line 376: “initial increase was attributed to TYL and immobilized carriers” please state your thesis more clearly. How could TYL and the immobilized carrier contribute to the increase in COD?

Lines 376-377: “whereas the decrease was attributed to degradation of TYL” what was the COD equivalent of TYL contained in wastewater?

Line 380: “probably due to bacteria utilizing P” Why was there a difference compared to the control?

Lines 380-382: not a logical explanation. First, ammonia levels generally do not increase during denitrification. Secondly, this is an aerobic process that does not favor denitrification.

General comment:

What about the safety concerns associated with strong pathogenicity of Klebsiella pneumonia? How does this relate to practical applicability? The propagation of an antibiotic-resistant pathogen is unlikely to be approved by the relevant authorities?

Reviewer #2: The review of the manuscript PONE-D-23-42913 has been completed. Concerns below needs to be addressed.

The abstract needs to be re-written to reflect the significance of the study.

Previous researches have reported the use of bacteria for the remediation of pollutant species. Authors should add some of these papers include their degradation efficiency

The study needs to be comprehensive. Adsorption accelerated oxidation and biodegradation of pollutant specie besides antibiotics should be added. Here are some references: https://doi.org/10.3390/ijms231810637, https://doi.org/10.3390/ijerph19169962, https://doi.org/10.3390/w14132063.

The novelty and significance of the study should be comprehensively presented.

6. PLOS authors have the option to publish the peer review history of their article (what does this mean?). If published, this will include your full peer review and any attached files.

Reviewer #1: No

Reviewer #2: No

---

## [Author Response · Author response to Decision Letter 0]

25 Mar 2024

respond to specific reviewer and editor comments：Response to Reviewers，Revised Manuscript with Track Changes，Manuscript，all three documents have been uploaded

---

## [Decision Letter · Decision Letter 1]

5 Apr 2024

PONE-D-23-42913R1Development and assessment of an immobilized bacterial alliance that efficiently degrades tylosin in wastewaterPLOS ONE

Dear Dr. Gao,

Thank you for submitting your manuscript to PLOS ONE. After careful consideration, we feel that it has merit but does not fully meet PLOS ONE’s publication criteria as it currently stands. Therefore, we invite you to submit a revised version of the manuscript that addresses the points raised during the review process.

The reviewers' comments need to be addressed, and the manuscript must be revised accordingly before it can be accepted for publication.

We look forward to receiving your revised manuscript.

Kind regards,

Marcos Pileggi, Ph.D

Academic Editor

PLOS ONE

Journal Requirements:

Reviewers' comments:

Reviewer's Responses to Questions

**Comments to the Author**

1. If the authors have adequately addressed your comments raised in a previous round of review and you feel that this manuscript is now acceptable for publication, you may indicate that here to bypass the “Comments to the Author” section, enter your conflict of interest statement in the “Confidential to Editor” section, and submit your "Accept" recommendation.

Reviewer #1: All comments have been addressed

Reviewer #2: All comments have been addressed

2. Is the manuscript technically sound, and do the data support the conclusions?

Reviewer #1: Yes

Reviewer #2: Yes

3. Has the statistical analysis been performed appropriately and rigorously? 

Reviewer #1: Yes

Reviewer #2: N/A

4. Have the authors made all data underlying the findings in their manuscript fully available?

Reviewer #1: Yes

Reviewer #2: Yes

5. Is the manuscript presented in an intelligible fashion and written in standard English?

Reviewer #1: Yes

Reviewer #2: Yes

6. Review Comments to the Author

Reviewer #1: Thank you for addressing my comments.

Please

1) Lines 82-84: You should not give description of agricultural environment in the aim of the work. It can be indicated in the introduction above.

2) Lines 275-278: Just define SEM in parentheses rather than in a separate sentence, it's redundant.

3) Line 298: pH, not PH

4) Be clear about whether you are working with aquaculture or agricultural wastewater?

5) Lines 369-375: Do you mean that composite carbon is sourced from beads and those beads are being biodegraded? Because it seems that wastewater did not have undissolved organics.

Reviewer #2: The review of the revised manuscript titler'Development

and assessment of an immobilised bacterial alliance that effectively degrades tylosin in wastewater has been completed. The concern below is required to be addressed before manuscript can be accepted.

The manuscript has been improved comprehensively. However, before it can be accepted. Authors should address the minor comment below.

The Line 79-80 should be supported with a hypothesis to give deeper insight into the objective of the study.

7. PLOS authors have the option to publish the peer review history of their article (what does this mean?). If published, this will include your full peer review and any attached files.

Reviewer #1: No

Reviewer #2: **Yes: **Abdulrahman Oyekanmi Adeleke

---

## [Author Response · Author response to Decision Letter 1]

10 Apr 2024

Response to Reviewers，Revised Manuscript with Track Changes，Manuscript，All three manuscripts have been revised in accordance with the comments

---

## [Decision Letter · Decision Letter 2]

7 May 2024

Development and assessment of an immobilized bacterial alliance that efficiently degrades tylosin in wastewater

PONE-D-23-42913R2

Dear Dr. Gao,

We’re pleased to inform you that your manuscript has been judged scientifically suitable for publication and will be formally accepted for publication once it meets all outstanding technical requirements.

Kind regards,

Marcos Pileggi, Ph.D

Academic Editor

PLOS ONE

Additional Editor Comments (optional):

Reviewers' comments:

Reviewer's Responses to Questions

**Comments to the Author**

1. If the authors have adequately addressed your comments raised in a previous round of review and you feel that this manuscript is now acceptable for publication, you may indicate that here to bypass the “Comments to the Author” section, enter your conflict of interest statement in the “Confidential to Editor” section, and submit your "Accept" recommendation.

Reviewer #1: (No Response)

Reviewer #2: All comments have been addressed

2. Is the manuscript technically sound, and do the data support the conclusions?

Reviewer #1: (No Response)

Reviewer #2: Yes

3. Has the statistical analysis been performed appropriately and rigorously? 

Reviewer #1: (No Response)

Reviewer #2: Yes

4. Have the authors made all data underlying the findings in their manuscript fully available?

Reviewer #1: (No Response)

Reviewer #2: Yes

5. Is the manuscript presented in an intelligible fashion and written in standard English?

Reviewer #1: (No Response)

Reviewer #2: Yes

6. Review Comments to the Author

Reviewer #1: (No Response)

Reviewer #2: Concerns has been comprehensively addressed. Manuscript is suitable for acceptance in the current form.

7. PLOS authors have the option to publish the peer review history of their article (what does this mean?). If published, this will include your full peer review and any attached files.

Reviewer #1: No

Reviewer #2: No

---

## [Editor Report · Acceptance letter]

20 May 2024

PONE-D-23-42913R2 

PLOS ONE

Dear Dr. Gao, 

I'm pleased to inform you that your manuscript has been deemed suitable for publication in PLOS ONE. Congratulations! Your manuscript is now being handed over to our production team.

Kind regards, 

on behalf of

Dr. Marcos Pileggi 

Academic Editor

PLOS ONE